# Diversity and relative abundance of bird species in the two habitat types of Dokima forest Awi zone, Ethiopia

**Binega Derebe Asmare** [1]*, **Yonas Derebe**[2], **Mulugeta Tamer**[2]

**1** Department of Natural Resource Management, College of Agriculture, Food and Climate Science, Injibara University, Gondar, Ethiopia, **2** Department of Forest and Climate Science, College of Agriculture, Food and Climate Science, Injibara University, Gondar, Ethiopia

\* binegaderebe@gmail.com

**Data Availability Statement:** Data are within the Supporting Information file.

**Funding:** Funding was provided by Injibara University, Ethiopia. The funders had no role in the

## Abstract

Birds are the most widespread vertebrate group, as they can be found in practically every type of habitat. However, lack of knowledge on bird ecology in Afrotropical highlands and bird habitat degradation are the main problems that is why this research was studied. The study was conducted in Awi zone, Amhara National Regional State with the objective of the diversity and relative abundance of bird species in the two habitat types of Dokima forest in Awi zone, Ethiopia. Between November 2018 and December 2020, the transect method was conducted in both dry and wet seasons. Using one-way ANOVA, the effect of seasons and habitats on species richness and abundance was investigated and compared. In both the dry and wet seasons, a total of 2233 individuals, 47 species belonging to 35 families, and 14 orders were recorded. The abundance of bird species was not statistically differed between habitat types in the dry season, but it was statistically significantly in the wet season. The mean abundance of bird species differed significantly between the dry and wet seasons. In the dry season, the forest habitat type had the highest species diversity index (H' = 3.18) and the highest evenness (J = 0.94), while in the wet season, the forest habitat type had the highest evenness (J = 0.94). These habitats must be conserved in order to protect the birds that live in the area.

## Introduction

It is important to note that afro-tropical forests are very rich in species and endemism and play an important role in biodiversity around the world [1]. As a result of their conspicuousness and ease of determination relative to other taxa, birds are significant organisms for evaluating the long-term impacts of human pressure on biodiversity in the tropics [2]. A bird can also be used as an indicator of biodiversity since they display sensitivity to various forms of human intervention [3]. Birds are taken as good indicators of biodiversity and monitors of environmental changes, like the level of contamination and environmental impacts [4]. Forest sites with higher woodpecker richness were also rich in all other bird species [5]. Birds eat pests,

study design, data collection, analysis, decision to publish, or preparation of the manuscript.

**Competing interests:** The authors have declared that no competing interests exist.

pollinate flowers, disseminate seeds, scavenge carrion, cycle nutrients, and alter the environment for the advantage of other species [6]. There are various organisms in situations where there are a big number of birds. Tropical inland lakes harbor and forest have a variety of birds and provide a wintering habitat for Palaearctic and other migratory birds [7]. Bird abundance and diversity can be used as indices or indicators of how ecologically diverse an ecosystem or habitat is [4]. There are several factors affecting bird populations in the tropics, including deforestation and forest fragmentation [8–11] and it leads to the particular ecological associations, such as insectivores and frugivores, are at risk [11–14]. Bird species abundance is determined by the distribution of food and cover resources. Species' requirements for food and cover are largely determined by the vegetation structure and composition, which are correlated with abundance and habitat use [15, 16]. Agricultural intensification, new agricultural commodities, residential growth, and land abandonment are all driving change in rural areas around the world [17]. Global biodiversity is threatened by deforestation in the humid tropics [18, 19]. It is predicted that global warming also cause elevational range shifts in bird species to montane habitats [19]. As humans encroach on forests in tropical areas, habitat loss and fragmentation impose further pressure on these avian communities [20–22]. Local, regional, and historical factors, such as competition, habitat variability, and climate changes, all influence bird species richness [23]. The nature of the flora that makes up a large part of a bird's habitat determines its species. Rainfall patterns that shift between wet and dry seasons are known to have an impact on vegetation composition and structure [15]. Biotic interactions and their dynamics influence species' relationships to climate, and this also has important implications for predicting future distributions of species [24]. Changes in human land use, such as grazing pressure and afforestation, have been cited as important threats to biodiversity in mountain environments, affecting species abundance and triggering distribution shifts to mountaintops [25]. Despite Ethiopia's high bird variety, habitat damage, fragmentation, and loss have been documented for decades, posing a serious threat to bird species' survival [26]. The habitat use, diversity studies and seasonal migratory of birds in Ethiopia is less explored [15]. Agricultural growth is a major cause of biodiversity loss, but the effects on agro-ecosystem community assembly are less well understood [27, 28]. Individual species must be considered in the context of the broader forest bird population when it comes to conservation. More shrub land birds will be supported by larger and more frequent clear-cuts, while older forest species may lose habitat. The abundance of shrub land birds varies with successional stage, depending on when optimal habitat conditions for a certain species arise [29].

Ethiopia is home to some of the world most unusual and diversified bird populations. Ethiopia is a huge, biologically varied country with a variety of unique environmental circumstances [4, 30]. Due to Ethiopia's topography, altitude and climate diversity, it is one of the few countries in the world with a high level of avian biodiversity [31]. One of Ethiopia's main reasons for its biodiversity richness is the imposing difference in altitude between Ras Dashen (4620 m above sea level) and Afar Depression (126 m below sea level) [32].There are 881 bird species in Ethiopia, including 19 endemics, 14 other bird species are shared with Eritrea, 31 globally threatened species, and one introduced species [33, 34]. Some birds have cultural values in Ethiopia, for instance Hornbill is a culturally significant bird species to the Oromo society where proverbs related to the species are used in constructing healthy social relationships [35]. A cautious interpretation of abundance and species richness data is also necessary, since deforestation is an extremely recent phenomenon, agricultural intensification is still occurring, and only a limited amount of information is available about the long-term stability of faunal populations in land use systems [18, 36]. Research into the species richness and distribution patterns of intact Afrotropical forests is necessary in order to fully understand disturbed ecosystems and communities [18]. Bird species richness and relative abundance and their role to

ecosystem functioning have been over looked by avian studies especially in developing countries like Ethiopia. As a result, the bird's check list of Ethiopia is still far from complete. Species composition, distribution, relative abundance and evenness of the bird fauna of Dokima forest are not before addressed. Therefore, the present study attempts to fill this gap. The objective of the study was to identify the diversity and relative abundance of bird species in the two habitat types of Dokima forest in Awi zone, Ethiopia.

## Materials and methods

### Study area description

Awi zone is one of the administrative zones in Amhara region located in between 11o to 10o85' N latitude and 36°39'60" to 36°57'E longitude [37]. According to Awi zone department of agriculture, 2018 report, most part of Awi zone is Woyena Dega (72%) followed by Dega (17%) and Kolla (11%). The area ranges from 700 to 2900m.asl in altitude [37]. According to the area's rainfall distribution, the dry season runs from December to April, while the wet season runs from May to November. The annual rainfall of the zone ranges between 1000 mm in the driest year to 1602 mm in the wettest year and the mean annual rainfall over those years is about 1302 with standard deviation of 110.4 mm [38, 39]. The temperature of the area ranges from 15 to 24oC. Awi zone is a home to a variety of wild animals' that include amphibians, reptiles, birds, and mammals. Dokima forest are one of the largest forest that found in Awi zone, it covers more than 2.5 hectares. Dokima forest is located in the Awi zone, in Banja woreda. It is situated between 10 58'3 0" N and 11 0' 0" N, and 36 38' 30"E and 36 40' 30"E, respectively.

### Study design and data collection

Dokima forest is characterized by moist Afro-montane forests with different vegetation strata including trees, shrubs and grasses [40]. The Dokima forest is a natural forest, no any plantation of trees. However the area is highly affected by anthropogenic impacts like logging, overgrazing and firewood collections. Some of the dominant tree species include *Cordia africana*, *Croton macrostachyus*, *Olea sepecies*, *Rosa abyssinica*, *Acacia abyssinica*, *Albizia species*, *Apodytes dimidiate*, *Ekebergia capensis*, *Ficus vasta*, *Prunus africana*, *Schefflera abyssinica*, *Rhus gluti* [40]. The sampling units representing each habitat types were selected based on stratified random sampling method. The area was divided into two habitat types for this study based on land cover features. Forest and shrub land were the habitat types under consideration. Bird species diversity and abundance were assessed using a random sampling design across the two stratified habitat categories. The censuses were taken on 30 sampling points of which 18 were from forest and 12 from shrubs. In each point count station, a minimum distance of 150–200m was maintained using GPS to avoid double counting [40].

Direct observations with the use of binoculars and bird guide books were used to identify birds and count individuals. Standing in the center of the point transects and softly observing up to a distance of 50 m radius was used to make observations. Each point transect was observed for 15 minutes [15]. Standing in the center of the point transects and silently and softly observing 360° about up to a radius of 50 m, observations were made [15]. The distance between points along the transect were 100m maximal in the shrub and 30m minimal in the forests [7]. The point count approach involves counting all individuals seen and heard by observers from a fixed location (census station) for a set period of time [7, 41].

During field observation, the birds' common and scientific names were recorded. To identify the bird species, the following three traits were used. External morphology (color, form, size, beak, leg, and tail), song and calls, and habitat type are the three factors to consider [42].

Point surveys of bird species were conducted in the morning from 6:00 to 10:00 a.m. and in the early evening from 5:00 to 7:30 p.m. [4, 43]. The survey point was visited two times a day in morning and afternoon and three days in wet and three days in dry season. Four observers with two data recorder were used during the survey. A prepared datasheet was used to record all of the bird species that were seen. By using simultaneous counting and thorough observation of birds while surveying birds, multiple counting of the same species or individual birds at a site was prevented. To obtain accurate data, well-experienced researchers and bird experts were involved with the aid of binocular and field guide books. Before conducting bird identification, all observers received introductory training on how to use the techniques and how to use field materials and tools [43].

## Data analysis

During the study period, all data was summarized in a table by season and habitat type. The Shannon-Wiener Diversity Index was used to calculate the distribution, abundance, and evenness of species across the wet and dry seasons, as well as between habitat types. The statistical analysis was performed using SPSS version 20 software and R studio. Using one-way ANOVA, the effect of seasons on species richness and abundance was investigated and compared. Excel version 2016 was used to generate the relative abundance and species diversity index using prepared formulas.

The following formula was used to calculate Shannon diversity index

$$H' = -\Sigma pi*\ln(pi) \dots\dots\dots\dots\dots\dots\dots\dots\dots\dots (1) \textbf{ Shannon diversity index}.$$

Where H' is Shannon-winner index, pi is estimated as ni/N, where ni is the proportion of the $i^{th}$ species and N = -Σni.

This use proportions rather than absolute abundance values to reduce the effects of order of magnitude deference in bird numbers between species. Birds' diversity was calculated using both Shannon-Weiner and Simpson's diversity indices [23, 43]. This index provides a measure of 'evenness' in the proportion of each species occurring within squares.

$$J' = H'/\ln(S) \dots\dots\dots\dots\dots\dots\dots (2) \text{ Evenness index}$$

Where, J' is Evenness index, H' is Shannon winner index and used the formula one and S is numbers of species encountered. S = Σn Where n is the number of species in a community

The similarity among and between the habitats concerning the composition of species was computed using Sorenson's similarity index (SI):

$$(SI) = 2C/S_1 + S_2 \dots\dots\dots\dots\dots\dots\dots (3) \text{ Similarity index}$$

Where C is the number of species the two habitats have in common, $S_1$ is the total number of species found in habitat 1, and S2 is the total number of species found in habitat 2.

$$\text{Relative abundance (RA) (\%)} = n/N \times 100 \dots\dots\dots\dots (4) \text{ Relative abundance}$$

Where, n is the number of individuals of particular species recorded and N is the total number of individuals of the species.

## Results

### Species richness and abundance of birds

In Dokima forest, a total of 2233 individuals of birds, 47 species belonging to 35 families and 14 orders, were identified throughout the study period (Table 1). In both the dry and wet

**Table 1. Bird species richness and abundance during both dry and wet seasons in Dokima forest.**

| Order of the species | Family | Scientific name | Common name | Population | | 2021 IUCN -CS |
|---|---|---|---|---|---|---|
| | | | | Dry | Wet | |
| Passeriformes | Estrildidae | *Uraeginthus bengalus* | Red-cheeked Cordon-bleu | 85 | 92 | LC |
| | | *Spermestes cucullata* | Bronze mannikin | 212 | 225 | LC |
| | | *Lagonosticta rufopicta* | Bare-breasted fire finch | 20 | 35 | LC |
| | Pycnonotidae | *Pychonotus barbatus* | Common bulbul | 13 | 22 | LC |
| | Oriolidae | *Oriolus monacha* | Ethiopian oriole [AB] | 20 | 22 | LC |
| | Coliidae | *Colius striatus* | Speckled mouse bird | 30 | 38 | LC |
| | Hirundinidae | *Cecropis daurica* | Red-rumped swallow | 25 | 36 | LC |
| | Ploceidae | *Bubalornis niger* | Red-billed buffalo weaver | 8 | 6 | LC |
| | | *Ploceus luteolus* | Little weaver | 23 | 20 | LC |
| | | *Ploceus intermedius* | Lesser masked weaver | 34 | 36 | LC |
| | | *Ploceus cucullatus* | Village weaver | 104 | 95 | LC |
| | Nectariniidae | *Cinnyris bifasciatus* | Purple-banded sunbird | 26 | 28 | LC |
| | Zosteropidae | *Zosterops senegalensis* | Yellow Wight eye sunbird | 8 | 9 | LC |
| | Muscicapidae | *Cossypha semirufa* | Rüppell's robin-chat | 45 | 55 | LC |
| | Turdidae | *Turdus smithi* | Karoo thrush | 29 | 39 | LC |
| | | *Zoothera piaggiae* | Abyssinian ground thrush | 10 | 12 | LC |
| | Malaconotidae | *Laniarius major* | Tropical boubou | 15 | 12 | LC |
| | Motacillidae | *Anthus cinnamomeus* | African pipit | 8 | 6 | LC |
| | Emberizidae | *Emberiza striolata* | Cinnamon-breasted bunting | 14 | 16 | LC |
| | Motacillidae | *Motacilla flava* | Western Yellow Wagtail | 6 | 4 | LC |
| | Viduidae | *Vidua chalybeate* | Village indigo bird | 4 | 8 | LC |
| | Fringillidae | *Crithagra gularis* | Streaky-headed seedeater | 62 | 62 | LC |
| | Monarchidae | *Terpsiphone viridis* | African paradise flycatcher | 10 | 8 | LC |
| | Alaudidae | *Mirafra cantillans* | Singing bush lark | 8 | 6 | LC |
| | Sturnidae | *Lamprotornis chalybaeus* | Greater blue-eared starling | 12 | 21 | LC |
| | Leiothrichidae | *Turdoides hartlaubii* | Hartlaub's babbler | 6 | 4 | LC |
| Coliiformes | Cisticolidae | *Phyllolais pulchella* | Buff-bellied warbler | 45 | 56 | LC |
| Coraciiformes | Coraciidae | *Eurystomus glaucurus* | Broad-billed roller | 8 | 10 | LC |
| | Meropidae | *Merops pusillus* | Little bee-eater | 56 | 66 | LC |
| Piciformes | Lybiidae | *Lybius bidentatus* | Double-toothed barbet | 8 | 6 | LC |
| | | *Lybius undatus* | Banded barbet [AB] | 4 | 8 | LC |
| | Picidae | *Dendropicos namaquus* | Bearded woodpecker | 6 | 4 | LC |
| Columbiformes | Columbidae | *Turtur abyssinicus* | Black-billed wood dove | 6 | 4 | LC |
| | | *Columba larvata* | Lemon dove | 8 | 12 | LC |
| | | *Streptopelia roseogrisea* | African collared dove | 2 | 8 | LC |
| | | *Spilopelia senegalensis* | Laughing dove | 6 | 4 | LC |
| Bucerotiformes | Bucerotidae | *Bycanistes brevis* | Silvery-cheeked hornbill | 4 | 8 | LC |
| Gruiformes | Rallidae | *Rougetius rougetii* | Rouget's rail [AB] | 2 | 2 | NT |
| Psittaciformes | Psittacidae | *Poicephalus flavifrons* | Yellow-fronted parrot | 8 | 6 | LC |
| Pelecaniformes | Threskiornithidae | *Bostrychia hagedash* | Hadada ibis | 4 | 6 | LC |
| Cuculiformes | Cuculidae | *Centropus monachus* | Blue-headed coucal | 4 | 4 | LC |
| Accipitriformes | Accipitridae | *Gyps rueppelli* | Rüppell's vulture | 4 | 2 | CR |
| | | *Melierax metabates* | Dark chanting goshawk | 6 | 4 | LC |
| Charadriiformes | Charadriidae | *Charadrius dubius* | Little ringed plover | 4 | 8 | LC |
| | | *Vanellus melanocephalus* | Spot-breasted lapwing [A] | 16 | 16 | LC |
| Musophagiformes | Musophagidae | *Turaco leucotis* | White-cheeked turaco | 8 | 12 | LC |

*(Continued)*

**Table 1.** (Continued)

| Order of the species | Family | Scientific name | Common name | Population | | 2021 IUCN -CS |
|---|---|---|---|---|---|---|
| | | | | Dry | Wet | |
| Passeriformes | Estrildidae | *Uraeginthus bengalus* | Red-cheeked Cordon-bleu | 85 | 92 | LC |
| Apodiformes | Apodidae | *Apus affinis* | Little swift | 10 | 14 | LC |

Note: A = Endemic[A]; AB = Endemic to both Ethiopia and Eritrea; NT = Near Threatened; LC = Least Concern; CR = Critically Endangered; CS = Conservation Status; Dry = Dry Season; Wet = Wet Season.

seasons, the order Passeriformes had the most species identified, followed by the Columbiformer (Fig 1). While the order of Bucerotiformes, Gruiformes, Pelecaniformes, and Cuculiformes was recorded in the study area had the least number of species (Fig 1). Passeriformes was also the most abundance in number of population follower by Coraciiformer (Fig 2). The correlation among the average populations and relative abundance were linear relationships, that means the higher the populations have the higher relative abundance (Fig 3). Ethiopian oriole (*Oriolus monacha*), Banded barbet (*Lybius undatus*), and Rouget's rail (*Rougetius rougetii*) are the only species found in the study area that are endemic to both Ethiopia and Eritrea. The Spot bearsted lawping (*Vanellus melanocephalus*), an endemic bird species, was also identified (Table 1). 45 species were classified as least concern by the IUCN in 2021, while two species, Rouget's rail (*Rougetius rougetii*) and Rüppell's vulture (*Gyps rueppelli*), were classified as near threatened and critically endangered, respectively (Table 1).

In the shrub habitat the average number of bird population was 1517±11.63 and in the forest habitat type the average number of bird population was 716±37.03. The average abundance of birds by habitat type was statistically significant different (P ≤0.001) (Table 2). Shrub habitat type was recorded the highest abundance of bird populations than forest habitat type. During the study 68% of bird population were recorded in shrub land and 32% of population of bird

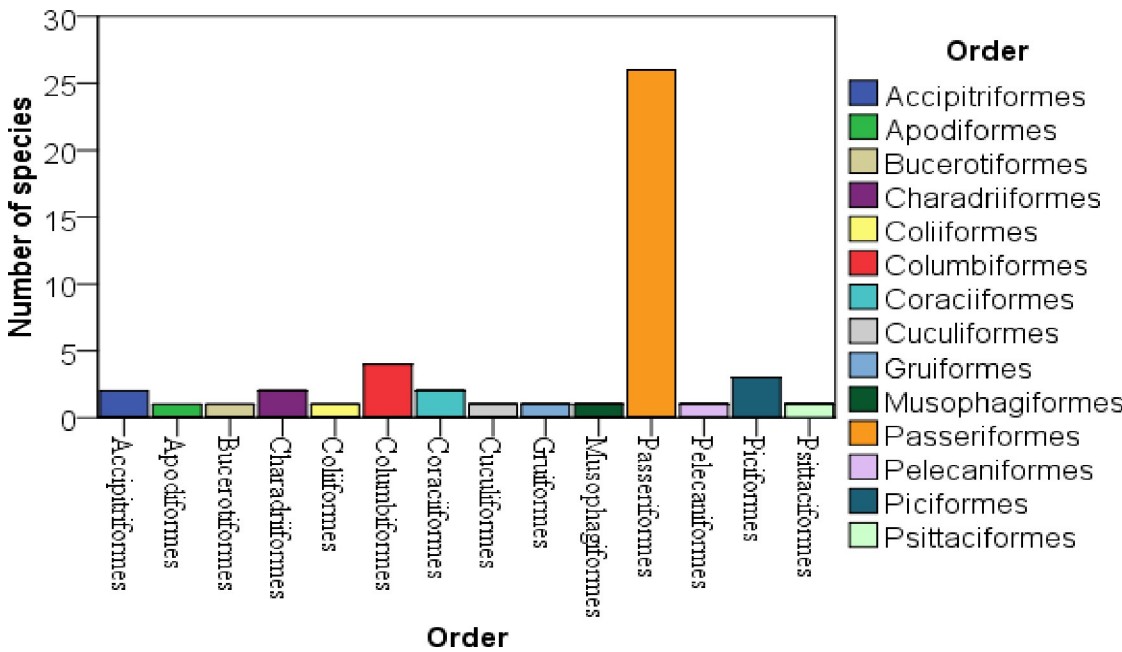

**Fig 1. Number of species among orders.**

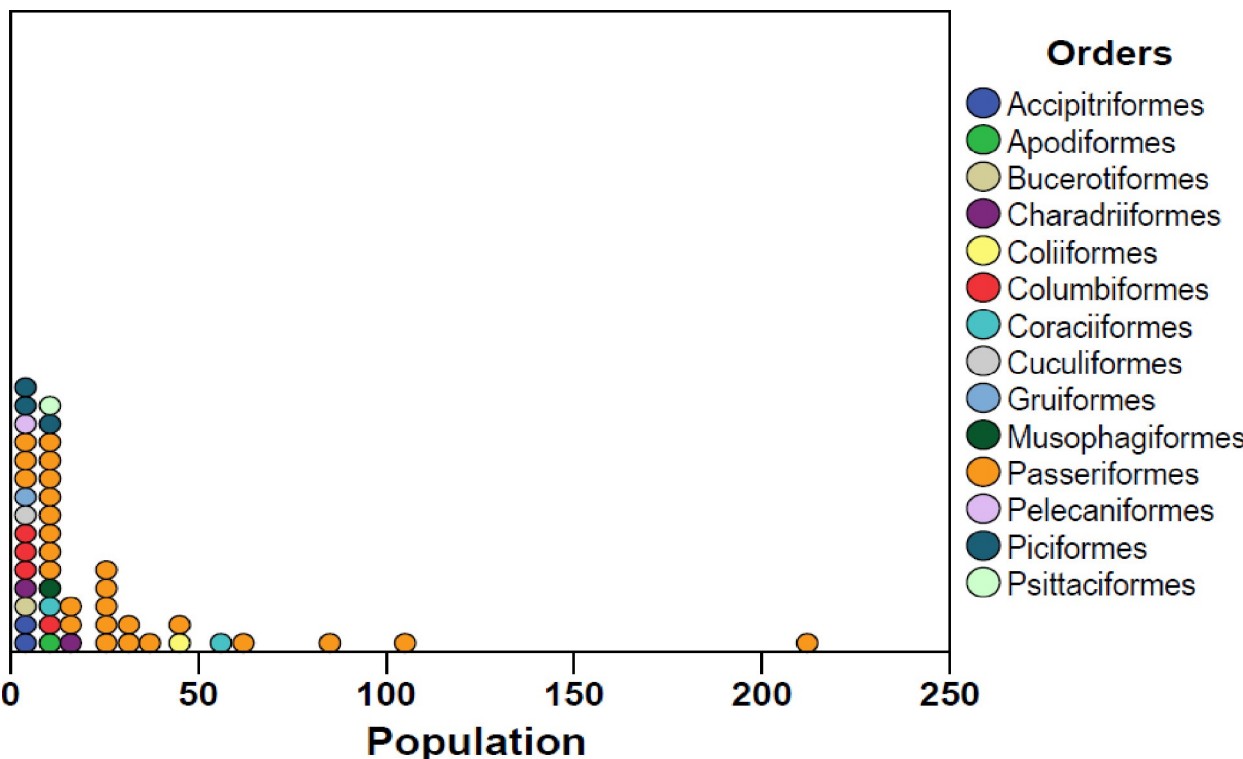

**Fig 2. Number of populations in each order.**

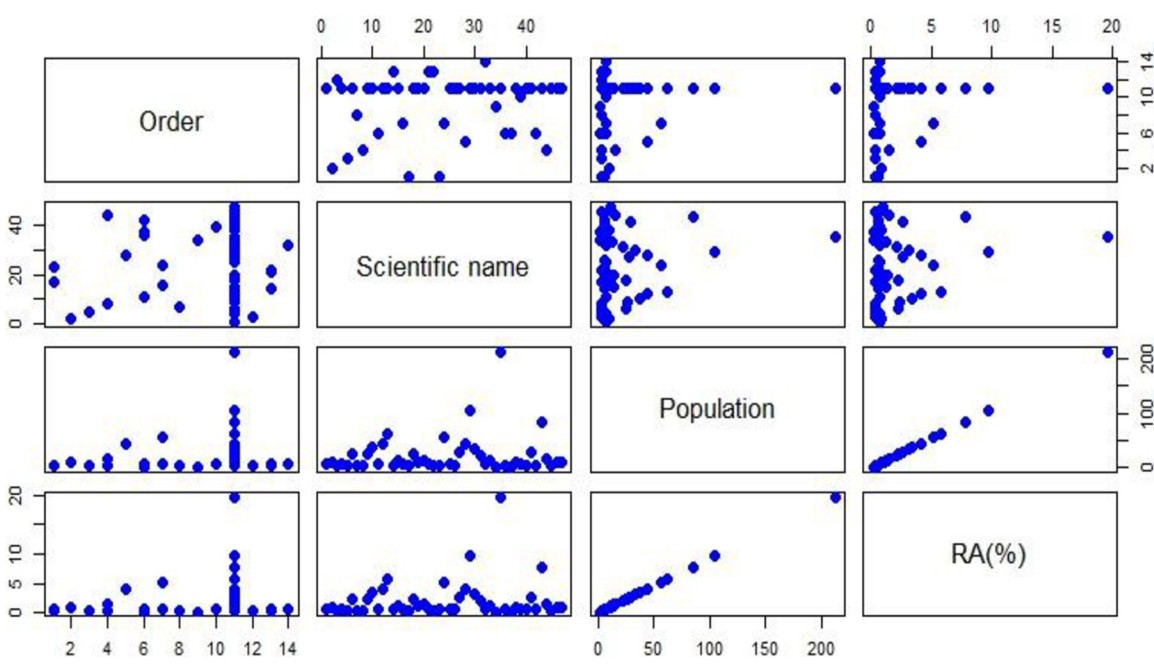

**Fig 3. Correlations among the entire matrix of four variables.**

**Table 2. Mean abundance of bird populations in the two habitats.**

| Habitat type | Mean number of populations | Std. De | Std. Error | Df | F | Sig. |
|---|---|---|---|---|---|---|
| Forest | 716 | 37.03 | 15.12 | 1 | 9298.56 | 0.000 |
| Shrub | 1517 | 11.63 | 4.75 | | | |

were recorded in forest habitat type (Fig 4). Forest habitat had the maximum species richness in both seasons 33 and 37 species in dry and wet respectively, while shrub land had the lowest species richness 30 and 31 species throughout the dry and wet seasons (Fig 5). However, Forest habitat type was higher number of species; whereas, its abundance was lower than shrub habitat type (344, 372) and (712, 805) individuals in both dry and wet seasons respectively (Fig 5).

Between the dry and wet seasons, there was a statistically significant variation in bird species abundance ($P \leq 0.001$). Wet season mean of bird abundance was higher than the mean abundance (1177±27.10.5) of dry season (1056 ± 7.3) (Table 3).

## Bird species diversity and evenness index

The species diversity of birds was recorded (H' = 3.18), (H = 3.4) in both dry and wet seasons, respectively. Although the highest species diversity was recorded in wet season, the diversity among season was almost the same. The evenness of the species distribution is lowest in the dry season (J = 0.366) and highest in the wet season (J = 0.94). In both the dry and wet seasons, the shrubs land habitat type reduces diversity, with the highest evenness index (J = 0.77) in the dry season (Table 4).

The Sorensen similarity index (S) is used to compare the similarity of species across various habitat types. It is set to "One" if two habitats are completely comparable, and "Zero" if the species of two habitat types are completely distinct. $S = 2C/(S_1 + S_2)$ is the Sorensen similarity index (S). During the dry seasons, the research area's forest habitat has 33 bird species, while the shrub land environment contains 30 and 16 bird species were common in both habitat types. During the wet season, there are 37 bird species in the forest habitat and 31 bird species in the shrub land environment and 21 bird species were common in both habitat types. In the

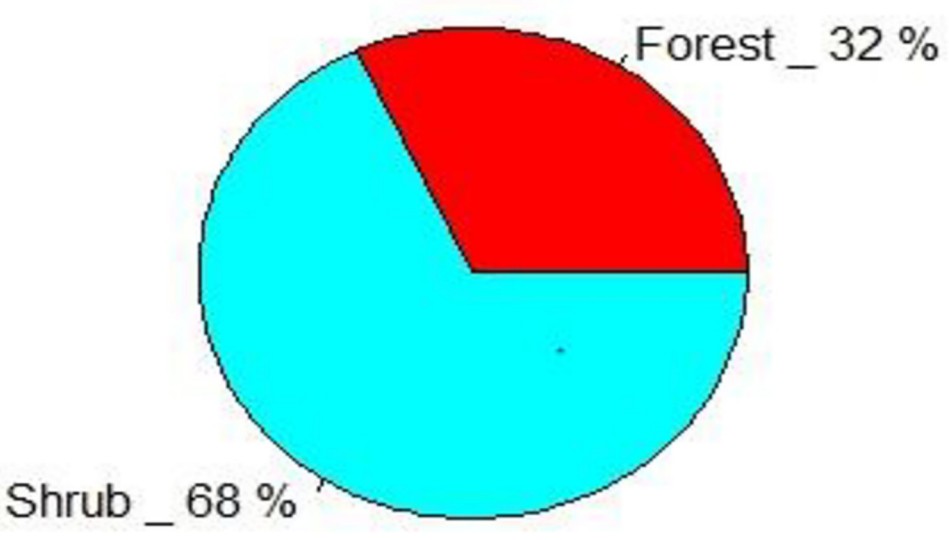

**Fig 4. Percentage of bird population among the two habitats.**

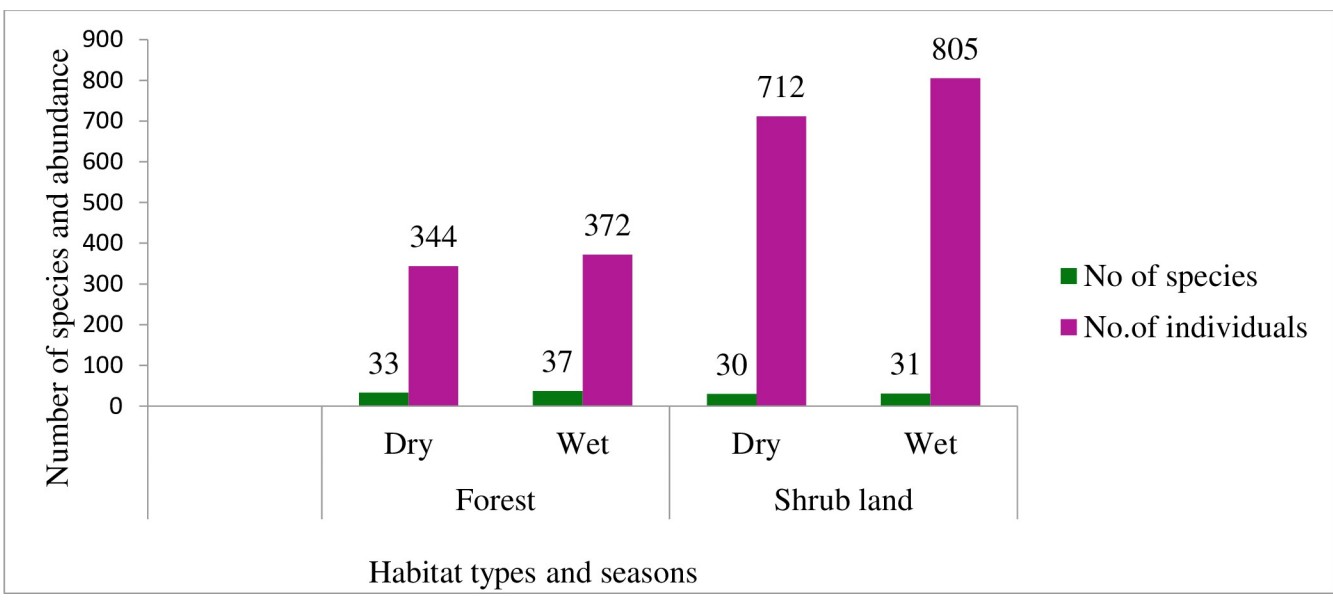

**Fig 5. Bird species abundance and distributions among the two study habitats.**

dry season, S = 2*16/ (33+30) = 32/63 = 0.51, whereas in the wet season, S = 2*21/ (37+31) = 42/68 = 0.61. This suggests that the species present in both environments in both seasons were more similar in wet season than the dry seasons.

## Relative abundance of birds

Bronze manikin (*Spermestes cucullata*) had the highest relative abundance (20.1 percent), (212 individuals), (19.1 percent), (225 individuals) in both the dry and wet seasons, followed by Village weaver (*Ploceus cucullatus*) with 104 (9.93 percent), 95 (8.07 percent) in both. During the dry season, the Roget rail (*Rougatius rougatti*) and African collared dove (*Streptopelia roseogrisea*) had the lowest relative abundance (2(0.2%) and 2(0.2%), respectively. The Roget rail (*Rougatius rougatti*) and the Ruppells vulture (*Gyps rueppelli*) had the lowest relative abundance score 2 (0.17 percent) each during the wet season (Table 5).

## Discussions

In comparison to other studies, the number of bird species in Dokima forest was low. For example, a total of 86 bird species were recorded in San Pedro de Puntina Punco [44]. In addition, 89 bird species were identified in the Zapotitlán Salinas Valle (Romero-Bautista et al., 2020). A total of 112 bird species belonging to 21 orders were recognized in Gibe Sheleko National Park (GSNP) [43] and 1,672 individuals belonging to 137 bird species were reported in Wondo Genet Forest, south-central Ethiopia [15]. In contrary, in the Buriganga river, a total of 38 bird species from 21 families and 8 orders were recorded [42]. Overall, effects on species richness were higher than those on bird abundance, with the latter being extremely

**Table 3. Total mean abundance of bird species in dry and wet season.**

| Seasons | Mean | Std. Deviation | Std. Error | Mini | Max | Df | F | Sig. |
|---|---|---|---|---|---|---|---|---|
| Dry season | 1056.000 | 23.16127 | 7.32424 | 1020.0 | 1092.00 | 1 | 88.78 | ≤0.001 |
| Wet season | 1177.000 | 33.35666 | 10.54830 | 1127.0 | 1227.00 | | | |

**Table 4. Bird species diversity and evenness index along habitat types.**

| Habitat type | Seasons | Species | No of individuals | Diversity(H') | Evenness (J) |
|---|---|---|---|---|---|
| Forest | Dry | 33 | 344 | 3.18 | 0.366 |
| | Wet | 37 | 372 | 3.4 | 0.94 |
| Shrubs | Dry | 30 | 712 | 2.62 | 0.77 |
| | Wet | 31 | 805 | 2.67 | 0.78 |

varied depending on the species planted and the geographical context in which these productive systems are found [45]. Species with a high degree of specialization are more vulnerable to environmental changes than generalist species with a larger geographic range [46].

Forest habitats had the maximum species richness in both dry and wet seasons, although it has lower bird population abundance than the shrub habitat type. This could be linked to the abundance of a range of food items, water, and cover during the research period, all of which contributed to the habitat's highest species richness and evenness [47]. With clear-cut age, bird communities shift from shrub land to mature forest, with more species associated with mature forest. Forest managers must address the geographical and temporal effects of timber harvests while managing forests for shrub land birds [29]. The high diversity and species richness of birds in the Dokima forest emphasizes the importance of implementing conservation measures and limiting human activity in the area according to different bird threat factors. For the sake of conserving high bird diversity in the entire riparian landscape, the wood cover, which includes trees, bushes, and young saplings, should be maintained, not only trees [48]. The forest patch and its environs are important for bird habitats [15]. Bird abundance and diversity are influenced by habitat type and size around the world, but especially in developing countries with rapid human population expansion and unplanned urban, agricultural, and industrial development [44]. The complexity of the ecosystem enhances the number of insects, which in turn promotes the diversity and population of birds [49]. Higher abundance during the wet season can be attributed to the availability of food as well as the breeding season [48]. All of the bird species found in the study area was sedentary species, they were found in both wet and dry seasons. Depending on the season, all regions were key bird habitats, reflecting the varying effects of temperature [50]. A similar research showed that the abundance of birds differed significantly between the two study seasons. The post-rainy season had the maximum abundance of birds, while the dry season had the lowest [7]. There was also a considerable change in the mean abundance of bird species between the dry and wet seasons in and around Wondo Genet woodland in south-central Ethiopia [15]. The diversity of bird species, on the other hand, did not vary significantly throughout the research period [47]. However, the species in our study varied according on the season.

During the study forest habitat was higher species diversity than the shrub habitat type. Forested areas have a higher diversity of forest-dependent bird species than non-forested areas [51]. Agricultural expansions in the study area had an impact on bird habitats; similarly, diverse land-use trajectories have resulted in variance in landscape structure, with a principal gradient of change from forest to rural townships [17]. Short-distance migrants and granivores birds are expected to increase as agriculture expands [28]. Similarly, despite conservation efforts in several bird-protected areas in Ethiopia, cropland has the lowest species diversity due to severe human disturbance [43]. Bird species richness was linked to environmental heterogeneity, habitat filtration, and biotic interactions [23]. The Rift Valley environment, which includes Lake Abijatta, has a total of 538 species of birds, accounting for more than 65 percent of the country's total [52]. Lakes with abundant vegetation gained more species than lakes with sparse vegetation, according to the relationship between habitat index and number of species

**Table 5. Relative abundance and distribution of bird species in the two study habitats with seasons.**

| Common name | Scientific name | RA (%) | | Habitat types | | | |
|---|---|---|---|---|---|---|---|
| | | Wet | Dry | FL | | SL | |
| | | | | Wet | Dry | Wet | Dry |
| Red-cheeked Cordon-bleu | *Uraeginthus bengalus* | 8.05 | 7.8 | - | - | + | + |
| Common bulbul | *Pychonotus barbatus* | 1.87 | 1.23 | + | + | + | + |
| Ethiopian oriole [NE] | *Oriolus monacha* | 1.97 | 1.79 | + | + | - | - |
| Village weaver | *Ploceus cucullatus* | 8.07 | 9.52 | + | + | + | + |
| Speckled mouse bird | *Colius striatus* | 3.23 | 2.55 | + | + | + | + |
| Buff-bellied warbler | *Phyllolais pulchella* | 4.76 | 4.26 | + | + | + | + |
| Little weaver | *Ploceus luteolus* | 1.7 | 2.18 | - | - | + | + |
| African paradise flycatcher | *Terpsiphone viridis* | 0.7 | 0.95 | + | + | + | + |
| Little bee- eater | *Merops pusillus* | 5.6 | 5.3 | + | + | + | + |
| Double-toothed barbet | *Lybius bidentatus* | 0.5 | 0.76 | + | + | + | + |
| Purple banded sunbird | *Cinnyris bifasciatus* | 2.38 | 2.46 | + | + | + | + |
| Yellow Wight eye sunbird | *Zosterops senegalensis* | 0.76 | 0.75 | + | + | + | + |
| Rüppell's robin-chat | *Cossypha semirufa* | 4.67 | 4.26 | + | + | + | + |
| Black billed dove | *Turtur abyssinicus* | 0.34 | 0.57 | + | + | + | - |
| Karoo thrush | *Turdus smithi* | 3.3 | 2.75 | + | + | + | + |
| Abyssinian ground thrush | *Zoothera piaggiae* | 1.02 | 0.95 | + | + | + | + |
| Tropical boubou | *Laniarius major* | 1.02 | 1.42 | + | + | - | - |
| Bare- breasted firefinch | *Lagonosticta rufopicta* | 3 | 1.71 | - | - | + | + |
| African pipit | *Anthus cinnamomeus* | 0.5 | 0.76 | + | + | - | - |
| Cinnamon-breasted bunting | *Emberiza striolata* | 1.34 | 1.33 | + | + | + | + |
| Laser masked weaver | *Ploceus intermedius* | 3.06 | 3.22 | + | + | + | + |
| Yellow wag tail | *Motacilla flava* | 0.34 | 0.57 | - | - | + | + |
| Bronze manikin | *Spermestes cucullata* | 19.1 | 20.1 | - | - | + | + |
| Lemon dove | *Columba larvata* | 1.02 | 0.76 | + | + | + | + |
| Village indigobird | *Vidua chalybeate* | 0.68 | 0.38 | - | - | + | + |
| Streaky seed eater | *Crithagra gularis* | 5.27 | 5.87 | + | + | + | + |
| Silvery-cheeked hornbill | *Bycanistes brevis* | 0.68 | 0.38 | + | + | - | - |
| Rouget's rail [NE] | *Rougatius rougatti* | 0.17 | 0.2 | - | - | + | + |
| Singing bush lark | *Mirafra cantillans* | 0.5 | 0.76 | - | - | + | + |
| Greater blue-eared starling | *Lamprotornis chalybaeus* | 1.78 | 1.14 | + | + | + | + |
| Broad-billed roller | *Eurystomus glaucurus* | 0.85 | 0.76 | + | + | - | - |
| Laughing dove | *Spilopelia senegalensis* | 0.34 | 0.57 | + | + | + | + |
| Yellow-fronted parrot | *Poicephalus flavifrons* | 0.5 | 0.76 | + | + | - | - |
| African collared dove | *Streptopelia roseogrisea* | 0.68 | 0.2 | + | + | - | - |
| Red-rumped swallow | *Cecropis daurica* | 3.06 | 2.37 | + | + | + | + |
| Hadada ibis | *Bostrychia hagedash* | 0.5 | 0.38 | - | - | + | + |
| Hartlaub's babbler | *Turdoides hartlaubii* | 0.34 | 0.57 | + | + | - | - |
| Blue-headed coucal | *Centropus monachus* | 0.34 | 0.38 | + | + | - | - |
| Rüppell's vulture | *Gyps rueppelli* | 0.17 | 0.38 | + | + | + | - |
| Red billed buffalo weaver | *Bubalornis niger* | 0.5 | 0.76 | - | - | + | + |
| Little ringed plover | *Charadrius dubius* | 0.68 | 0.38 | + | + | - | - |
| Banded barbet [NE] | *Lybius undatus* | 0.68 | 0.38 | + | + | - | - |
| Dark chanting goshawk | *Melierax metabates* | 0.34 | 0.57 | + | + | - | - |
| Spot-breasted lapwing [E] | *Vanellus melanocephalus* | 1.36 | 1.5 | + | + | - | - |
| White-cheeked turaco | *Turaco leucotis* | 1.53 | 1.02 | + | - | - | - |

*(Continued)*

**Table 5.** (Continued)

| Common name | Scientific name | RA (%) | | Habitat types | | | |
|---|---|---|---|---|---|---|---|
| | | Wet | Dry | FL | | SL | |
| | | | | Wet | Dry | Wet | Dry |
| Little swift | *Apus affinis* | 1.2 | 0.85 | + | - | + | - |
| Bearded woodpecker | *Dendropicos namaquus* | 0.34 | 0.51 | + | - | - | - |

**Note**: RA: Relative abundance, (+) refers to the species was found in the habitat and (-) refers to the species was not found in the habitat during the study, FL = Forest land, SL = Shrub land, Wet = Wet season, Dry = Dry season.

gained [53]. The forest patch and its environs are important bird habitats [15]. Anthropogenic changes to the landscape and climate result in new ecological and evolutionary constraints, potentially leading to major changes in biodiversity distribution [54]. Human-caused habitat loss and degradation are one of the most serious threats to biodiversity worldwide (Regos et al., 2018). For the long-term protection of bird communities, appropriate management programs must be designed and implemented [49].

There is some difference in bird populations among both dry and wet seasons as the relative abundance shows. The relative abundance of bird species during different seasons may be related to food availability, habitat conditions, and the species' breeding season. Seasonal fluctuations in bird species abundance are caused by the unique seasonality of rainfall and seasonal variation in the amount of food supplies [55, 56].

## Conclusion and recommendations

The habitats that were studied are significant for a variety of bird species. The presence of a large number of resident and international concerned birds in Dokima forest, including endemic and globally vulnerable species, proves the importance of the study areas for bird conservation. During the study, a large number of birds from both habitats were identified during the wet season. In all seasons, the forest habitat type had the greatest diversity of bird species. The mean abundance of bird species differed significantly between the dry and wet seasons. Bird diversity in shrub habitat was lower than in forest habitat. Bronze mannikin (*Spermestes cucullata*) had the largest percentage relative abundance of birds, followed by Village weaver (*Ploceus cucullatus*). During the study period, human disturbances such as livestock overgrazing, agricultural expansion, deforestation, and hunting were identified as the top threats to birds. To ensure bird conservation through their habitats, proper habitat conservation and management efforts should be implemented.

## Supporting information

**S1 Data. Dokima bird data.**
(XLSX)

## Acknowledgments

We like to say thank you for the College of Agriculture Food and Climate Science for all support. We owe a special appreciation to the experts and residents in the study area who provided us all the relevant information.

## Author Contributions

**Conceptualization:** Binega Derebe Asmare, Yonas Derebe.

**Data curation:** Binega Derebe Asmare, Yonas Derebe.

**Formal analysis:** Binega Derebe Asmare.

**Funding acquisition:** Yonas Derebe, Mulugeta Tamer.

**Investigation:** Binega Derebe Asmare.

**Methodology:** Binega Derebe Asmare.

**Project administration:** Binega Derebe Asmare.

**Resources:** Binega Derebe Asmare, Mulugeta Tamer.

**Software:** Binega Derebe Asmare.

**Supervision:** Binega Derebe Asmare, Mulugeta Tamer.

**Validation:** Binega Derebe Asmare.

**Visualization:** Binega Derebe Asmare, Yonas Derebe.

**Writing – original draft:** Binega Derebe Asmare.

**Writing – review & editing:** Binega Derebe Asmare.

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
