## [Decision Letter · Decision Letter 0]

1 Dec 2022

PONE-D-22-23950Diversity and relative abundance of bird species in the two habitat types of Dokima forest Awi zone, EthiopiaPLOS ONE

Dear Dr. Asmare,

Thank you for submitting your manuscript to PLOS ONE. After careful consideration, we feel that it has merit but does not fully meet PLOS ONE’s publication criteria as it currently stands. Therefore, we invite you to submit a revised version of the manuscript that addresses the points raised during the review process.

Although we have only one reviewer for your manuscript, they did a very good job in suggesting what is needed to improve the manuscript- please consider all their points seriously.  To those I would addIt appears that you have done three one-way ANOVAs- one for the dry season comparing the two habitats, one for the wet season comparing the two habitats and one for both habitats comparing the two seasons.  It is generally more appropriate to conduct a two-way ANOVA that has season and habitat as factors and includes an interaction term.  You may already have done this as from your results its likely that you would get a significant interaction term and you would then have to conduct separate tests.  However, this does bring you into the minefield of defining your factors as fixed vs random, nesting and the type of contrast tests to explore the different habitat/season differences.  Another way of looking at this is to determine whether you are most interested in season or habitat and divide the data and analyses based on that.  So if differences between habitats is your focus- then do an ANOVA that compares the habitats regardless of season and then show differences between seasons in each habitat separately with a plot (and then you could show your ANOVA results for the seasons separately).  Conversely if its differences between seasons- then show your ANOVA that compares the seasons regardless of habitat and then show differences between habitats in each season separately with a plot (and then you could show your ANOVA results for the habitats separately). Secondly your sorensons index.  You give two different equations for it and then after explaining that it ranges from 0 to 1 you give results that are greater than one.  I would suggest you either use the following equation, or use the jaccard index which is very similar but does not emphasise common species (just remove the 2) in the sorenson equationsorenson = (2a)/(2a + b + c) where a is the number of species in common, b is the number of species that only occur in the first sample and c is the number of species that only occur in the second sample Please submit your revised manuscript by Jan 15 2023 11:59PM. If you will need more time than this to complete your revisions, please reply to this message or contact the journal office at plosone@plos.org. Please include the following items when submitting your revised manuscript:A rebuttal letter that responds to each point raised by the academic editor and reviewer(s). You should upload this letter as a separate file labeled 'Response to Reviewers'.A marked-up copy of your manuscript that highlights changes made to the original version. You should upload this as a separate file labeled 'Revised Manuscript with Track Changes'.An unmarked version of your revised paper without tracked changes. You should upload this as a separate file labeled 'Manuscript'.

We look forward to receiving your revised manuscript.

Kind regards,

Judi Hewitt

Academic Editor

PLOS ONE

Journal Requirements:

2. In your Methods section, please provide additional information regarding the permits you obtained for the work. Please ensure you have included the full name of the authority that approved the field site access and, if no permits were required, a brief statement explaining why.\\

Injibara University was the funder

The data collection costs for this study were paid by Injibara University's College of Agriculture, Food and Climate Science.

However, funding information should not appear in the Acknowledgments section or other areas of your manuscript. We will only publish funding information present in the Funding Statement section of the online submission form. 

Injibara University was the funder

6. Thank you for stating the following in your Competing Interests section:  

No any conflict of interest 

7. We note that you have indicated that data from this study are available upon request. PLOS only allows data to be available upon request if there are legal or ethical restrictions on sharing data publicly. For more information on unacceptable data access restrictions, please see http://journals.plos.org/plosone/s/data-availability#loc-unacceptable-data-access-restrictions. 

8. We note that Figure 1 in your submission contain [map/satellite] images which may be copyrighted. All PLOS content is published under the Creative Commons Attribution License (CC BY 4.0), which means that the manuscript, images, and Supporting Information files will be freely available online, and any third party is permitted to access, download, copy, distribute, and use these materials in any way, even commercially, with proper attribution. For these reasons, we cannot publish previously copyrighted maps or satellite images created using proprietary data, such as Google software (Google Maps, Street View, and Earth). For more information, see our copyright guidelines: http://journals.plos.org/plosone/s/licenses-and-copyright.

Reviewers' comments:

Reviewer's Responses to Questions

**Comments to the Author**

1. Is the manuscript technically sound, and do the data support the conclusions?

Reviewer #1: Partly

2. Has the statistical analysis been performed appropriately and rigorously? 

Reviewer #1: I Don't Know

3. Have the authors made all data underlying the findings in their manuscript fully available?

Reviewer #1: Yes

4. Is the manuscript presented in an intelligible fashion and written in standard English?

Reviewer #1: No

5. Review Comments to the Author

Reviewer #1: This is an interesting work focuses on greatly understudied topic – abundances of birds in Afrotropical highlands. I applaud the authors for collecting nice data and composing this manuscript. At the same time, however, this manuscript suffers from several major weak points that need to be thoroughly addressed.

1. Writing style need considerable improvement. All parts are too wordy and often lack substantial information content. Here are specific comments to respective parts:

- Abstract: Please add one sentence why is this study important (e.g. lack of knowledge on bird ecology in Afrotropical highlands) and specify what are the focal habitat types right from the beginning. Please delete F-statistics, p-values and df.

- Introduction: There is a lot of redundant information, some of them are even repetitive (e.g. that about woodpecker richness). On the other hand, readers do not know from this text what are the aims of the study. So, I suggest building the Introduction as follows (the points roughly represent intended paragraphs): a) importance of Afrotropical forests for global biodiversity, b) threats to these forests and outcomes of these pressures (some parts are cleared and transformed to shrub land), c) importance of birds in ecological research, d) birds of Ethiopian highland forests + lack of knowledge on their abundance and species richness in different habitat types of this environment, e) study aims.

- Study area and study site: Please reduce the information on administrative divisions, state simply where the study was conducted and describe briefly the environmental conditions (climate, biotopes) in the area. All the information can be presented in 5-8 sentences. The justification for selecting the study area for research should be moved to the Introduction.

- Study design and data collection: Please state briefly the sampling period and remove the information about the preliminary survey as it is irrelevant. Description of the habitats should be moved to the previous chapter. Description of bird counts must be elaborated since some sentences are duplicated (but differ in the specific information – see 30m vs. 50m radius!). Many of the information are redundant (e.g. readers do not need to know your bird identification techniques or how did you make your notes on bird detections). When writing this part, please have in mind that it serves for potential reproduction of your research by someone, not for justification of your bird identification skills. Consider the presentation from this perspective.

- Data analysis: Please describe in more detail the ANOVA. You state that it was used for testing the differences between seasons, but you obviously used it to test for differences between habitats, too. Moreover, the Sorenson similarity index cannot work in the form presented in the text. It is defined as a proportion of species common to two assemblages. So, it cannot be calculated as a number of species common to both habitats divided by the sum of the total numbers of species recorded in respective habitats. If both habitats contained the same 10 species, then it would be 10/(10+10) = 0.5, whereas the correct value of this index should be 1.0. Please revise the formula.

- Results and Discussion: I do not see any reason for merging these two parts. It makes the text very unclear and reduces the true discussion to minimum. Please separate results to a specific Results section (its organization into subchapters can be the same as it is) and then write a new Discussion section. In Discussion, we need to learn the interpretation of the patterns presented in the Results section, i.e. why there were the differences between the habitats and seasons observed. In addition, please make a comparison with other studies from Afrotropical region in general and from the mountain ecosystems in particular.

2. Conducting the research. Based on the information presented, it seems that you used mainly visual detections. This is quite unusual as 90%+ birds are recorded as aural detections in forest conditions. At the same time, you state that birds were recorded by experts and experiences researchers, so it sounds odd that “binoculars and bird guide books were used” – experts just use their knowledge of bird voices. Moreover, I do not know what was the radius used for bird counts – 30m or 50m? Both figures mean that the radii overlapped in forest habitat because points were separated by 30m distance. Finally, we do not know how the distances were measured – by eye after training or by laser range finder? Taken together, the description of data collection must be much more convincing to judge reliability of the data.

3. Data analysis. I think that one-way ANOVA is an appropriate and powerful tool for addressing the questions you asked, but we do not know what the data points are, and what the explanatory variables are. The other analyses are less convincing. The correlations shown in Fig. 4 make little sense, especially those between population and relative abundance – these variables are identical, they only differ in the way of presentation of the same thing. So, their correlation must be 1.0, as also indicated by the plot. Values of the Sorenson index do not correspond to the theoretical background for this index and are most likely wrong.

4. Comparison with other studies. The data on bird abundance remain scarce in Afrotropical highland conditions, but some studies exist – please consult literature from Albertine rift, Mt. Kenya, Kilimanjaro and our work from the Cameroon Mountains to make a thorough assessment of your findings in the context of other studies.

Jiri Reif

6. PLOS authors have the option to publish the peer review history of their article (what does this mean?). If published, this will include your full peer review and any attached files.

Reviewer #1: **Yes: **Jiri Reif

---

## [Author Response · Author response to Decision Letter 0]

14 Dec 2022

Dear reviewers, I would like to say thank you for your best and constructive comments. We try to to correct the comments but some suggestions were difficult to us like Map of the study area copyright and Data depository because especially in data depository, the data is in my hand but how I send the data to the data depositors. Thank you again!!

Best regards!!

---

## [Editor Report · Decision Letter 1]

27 Jan 2023

Diversity and relative abundance of bird species in the two habitat types of Dokima forest Awi zone, Ethiopia

PONE-D-22-23950R1

Dear Dr. Asmare,

We’re pleased to inform you that your manuscript has been judged scientifically suitable for publication and will be formally accepted for publication once it meets all outstanding technical requirements.

Kind regards,

Judi Hewitt

Academic Editor

PLOS ONE
---

## [Editor Report · Acceptance letter]

31 Jan 2023

PONE-D-22-23950R1 

Diversity and relative abundance of bird species in the two habitat types of Dokima forest Awi zone, Ethiopia 

Dear Dr. Asmare:

I'm pleased to inform you that your manuscript has been deemed suitable for publication in PLOS ONE. Congratulations! Your manuscript is now with our production department. 

Kind regards, 

on behalf of

Dr. Judi Hewitt 

Academic Editor

PLOS ONE